## REVIEW ARTICLE

# Perspectives of glycemic variability in diabetic neuropathy: a comprehensive review

Xiaochun Zhang[1,5], Xue Yang[2,5], Bao Sun [3,4✉] & Chunsheng Zhu [1✉]

Diabetic neuropathy is one of the most prevalent chronic complications of diabetes, and up to half of diabetic patients will develop diabetic neuropathy during their disease course. Notably, emerging evidence suggests that glycemic variability is associated with the pathogenesis of diabetic complications and has emerged as a possible independent risk factor for diabetic neuropathy. In this review, we describe the commonly used metrics for evaluating glycemic variability in clinical practice and summarize the role and related mechanisms of glycemic variability in diabetic neuropathy, including cardiovascular autonomic neuropathy, diabetic peripheral neuropathy and cognitive impairment. In addition, we also address the potential pharmacological and non-pharmacological treatment methods for diabetic neuropathy, aiming to provide ideas for the treatment of diabetic neuropathy.

With the improvement of people's living standard and the increase of competitive pressure, there is growing number of patients with diabetes and diabetes-related complications[1]. Diabetic neuropathy (DN) is among the most common long-term complications of diabetes, with significant morbidity and mortality. It includes both peripheral and autonomic neuropathy, and is estimated to affect more than 60% of diabetes patients[2].

Although chronic hyperglycemia is traditionally considered as a major risk factor for diabetes-related complications, it has been suggested that frequent or large glucose fluctuations may independently lead to diabetes-related complications. In addition to hemoglobin A1c (HbA1c), glycemic variability (GV) could be another independent risk factor for diabetic complications[3]. Several large-scale clinical studies had identified that the greater degree of GV was significantly associated with the higher incidence of chronic complications of diabetes[4]. As for instance, an extensive HbA1c control cohort including 38 patients with type 2 diabetes mellitus (T2DM) identified that high GV was harmful to DN even in the context of normal HbA1c levels[5]. In recent years, GV has been paid extensive attention as an indicator to evaluate blood glucose control. Furthermore, GV, defined as the degree of blood glucose fluctuation and rarely caused by a single factor, was regarded as a potential independent risk factor for diabetic comlications[3,6–8]. Similarly, findings from studies in T2DM supported that there was a significant positive association between GV and the development or progression of diabetic retinopathy[9,10], cardiovascular events, and mortality[11–13]. Notably, GV tended to be a better glycemic parameter for assessing the risk of future micro- and macro-vascular complications in patients with T2DM[3,14,15].

Here, we elaborate the role and related mechanisms of GV in DN, including cardiovascular autonomic neuropathy (CAN), diabetic peripheral neuropathy (DPN), and cognitive impairment. In parallel, we also discuss the potential pharmacological and non-pharmacological

[1]The First Affiliated Hospital of Zhengzhou University, Zhengzhou 450052, China. [2]CAS Key Laboratory of Separatin Science for Analytical Chemistry, National Chromatographic Research and Analysis Center, Dalian Institute of Chemical Physics, Chinese Academy Sciences, Dalian 116000, China. [3]Department of Pharmacy, The Second Xiangya Hospital, Central South University, Changsha 410000, China. [4]Institute of Clinical Pharmacology, Central South University, Changsha 410000, China. [5]These authors contributed equally: Xiaochun Zhang, Xue Yang. ✉email: scy_csu2016@csu.edu.cn; zhuchunsheng6@163.com

treatment methods for GV, aiming to provide new strategies for the treatment of diabetes with DN.

## Assessment of GV

Patients with diabetes face a life-long optimization problem of how to lower average blood sugar levels and postprandial hyperglycemia without causing hypoglycemia[16]. In the past decade, along with HbA1c, GV has been increasingly regarded as a primary marker of glycemic control[17–19]. With increasing interest in the importance of GV, a number of indicators have been proposed to characterize GV in clinical trials. The coefficient of variation (CV) and standard deviation (SD) are those adopted in the consensus of continuous glucose monitoring (CGM) indices of GV[16,20]. Some other currently used indices include mean amplitude of glycemic excursion (MAGE), continuous overall net glycemic action, mean of daily differences, high blood glucose index, low blood glucose index, glycemic risk assessment in diabetes equation[16,21–27]. Nonetheless, currently, there is little consensus on the standard method to assess GV. Notably, the Advanced Technologies & Treatments for Diabetes International Consensus recommends the use of CV to assess GV with a cutoff value of 36% in clinical practice[28].

In addition to self-blood glucose monitoring, that mainly supports self-management and medication adjustment of diabetic patients, here we emphasize CGM[29]. HbA1c, which reflects overall glycemic control over the first 60–90 days, has been considered the gold standard for assessing the outcome of diabetes management since 1993[30]. However, there is increasing recognition of the limitations of HbA1c as glucose control, due to the ignorance of fluctuation in blood glucose levels known as GV. The development of CGM systems has improved the analysis and interpretation of GV[31]. CGM provides detailed information on several aspects of glucose control, including GV[32]. Of note, CGM can reliably detect potential postprandial hyperglycemia with normal HbA1c level[33]. Furthermore, CGM can also systematically record daily glucose levels, making the data more representative without interfering with normal daily life[34]. Recently, CGM has been proven to be a useful indicator of GV in preclinical type 1 diabetes mellitus (T1DM)[35,36]. Taken together, CGM system has made it possible to accurately measure short-term GV and to investigate the role of glucose fluctuations in the development of diabetes-related complications[3].

It was reported that these indices of GV were largely correlated with each other[37,38]. For example, a previous study enrolling 88 Japanese patients with diabetes mellitus revealed that the GV indices, including index of glycemic control, mean of daily differences, continuous overall net glycemic action, and MAGE, obtained by CGM were closely correlated with SD glucose[39]. Conversely, more recently, a retrospective review found that daily GV and visit-to-visit GV was differently correlated with clinical parameters, and there was almost no connection between them[40]. Hence, further analysis is necessary to clarify the relationships among indices of GV.

## Roles of GV in DN

According to clinical reports, patients with diabetes will develop several types of DN damage, including CAN, DPN, and cognitive impairment[31,41,42]. Presently, the pathogenesis of DN is complicated, and possible mechanisms can be categorized as follows: oxidative stress, mitochondrial dysfunction, advanced level of glycation endproducts, polyol pathway, hexosamine, and protein kinase C pathways, etc[43] (Fig. 1). Moreover, accumulating evidence has revealed that the dysfunction of Schwann cells plays a significant role in the pathogenesis of DPN, such as apoptosis, lipid metabolism abnormality, oxidative stress, inflammatory reactions, and endoplasmic reticulum stresss[44,45]. In parallel,

there is growing evidence supporting that GV has drawn a great attention for its role in CAN, DPN, and cognitive impairment.

## GV and CAN

It is well known that there is bidirectional regulation between autonomic nervous activity and glucose metabolism[46,47]. Autonomic imbalance was prevalent and might develop to diabetic autonomic neuropathy in patients with diabetes[48]. Of note, recent evidence suggested that GV was involved in CAN with T1DM. A pilot study enrolling 44 T1DM patients from the University of Michigan Health System identified that the indices of GV reflective of hypoglycemic stress, low blood glucose index, and area under the curve for hypoglycemia, were significantly negative correlated with the low and high-frequency power of heart rate variability (CAN indicator), suggesting that GV was likely to contribute to CAN[49]. Similarly, Nyiraty et al. revealed that GV marker calculated by SD and mean absolute glucose were associated with the severity of CAN in patients with T1DM[50]. Nevertheless, in a cross-sectional observational study including 133 young adults with T1DM from 18 to 24 years, Christensen et al. reported that after adjusting for risk factors and multiple tests, only higher MAGE was associated with slightly increasing measures of heart rate variability, indicating that GV might not be a risk factor for CAN in young adults with T1DM[51].

Furthermore, reports have indicated that GV are considered important risk factors for CAN in subjects with T2DM. For instance, a previous study reported that the fluctuation of fasting sympathetic nerve activity around wake-up assessed by heart rate variability was positively correlated with short-term GV in T2DM patients[46]. Likewise, a Korea prospective study showed that short and long-term GV, such as CGM-SD, CGM-CV, SD of HbA1c and log CV of HbA1c etc, had significantly higher association with the presence of CAN in patients with T2DM than in those without T2DM, indicating that GV was independently associated with the presence of CAN in patients with T2DM[52]. In parallel, a retrospective cohort study including 681 subjects with T2DM reported that CAN was significantly associated with the risk of developing higher HbA1c variability measured by SD[53]. Moreover, baroreflex sensitivity, as a sensitive indicator of CAN in T2DM, was found to be inversely related to long-term GV represented by visit-to-visit HbA1c variability in patients with T2DM[54]. Analogously, Lai et al. showed that the HbA1c variability measured by SD was not only strongly related to the presence but also to the severity of CAN[55]. However, it was worth to note that in a non-insulin-treated T2DM cohort study consisting of 39 women and 48 men, the indicators of CAN including the standard deviation of normal-to-normal intervals, the root mean square of successive differences, total power, and expiration-to-inspiration ratio etc, were significantly correlated with the increase in MAGE in only women, implying that GV have gender-specific effects on CAN in patients with T2DM[56] (Table 1). As a consequence, further prospective studies are needed to confirm the role of GV and CAN in both T1DM and T2DM.

Limited by the current knowledge, the pathogenesis of CAN is not fully understood. However, emerging evidence suggested that CAN might be caused by changes in GV due to inflammatory reactions and oxidative stress (Fig. 1a)[57–59]. On the one hand, autonomic dysfunction occurred in the early stages of diabetes, which might be accompanied by changes in various inflammatory cytokines, including interleukin-6[57]. On the other hand, previous findings from in vitro studies showed that acute blood glucose fluctuations induced a greater trigger effect on oxidative stress through reactive oxygen species overproduction at the mitochondrial electron transport chain[60]. Moreover, increased

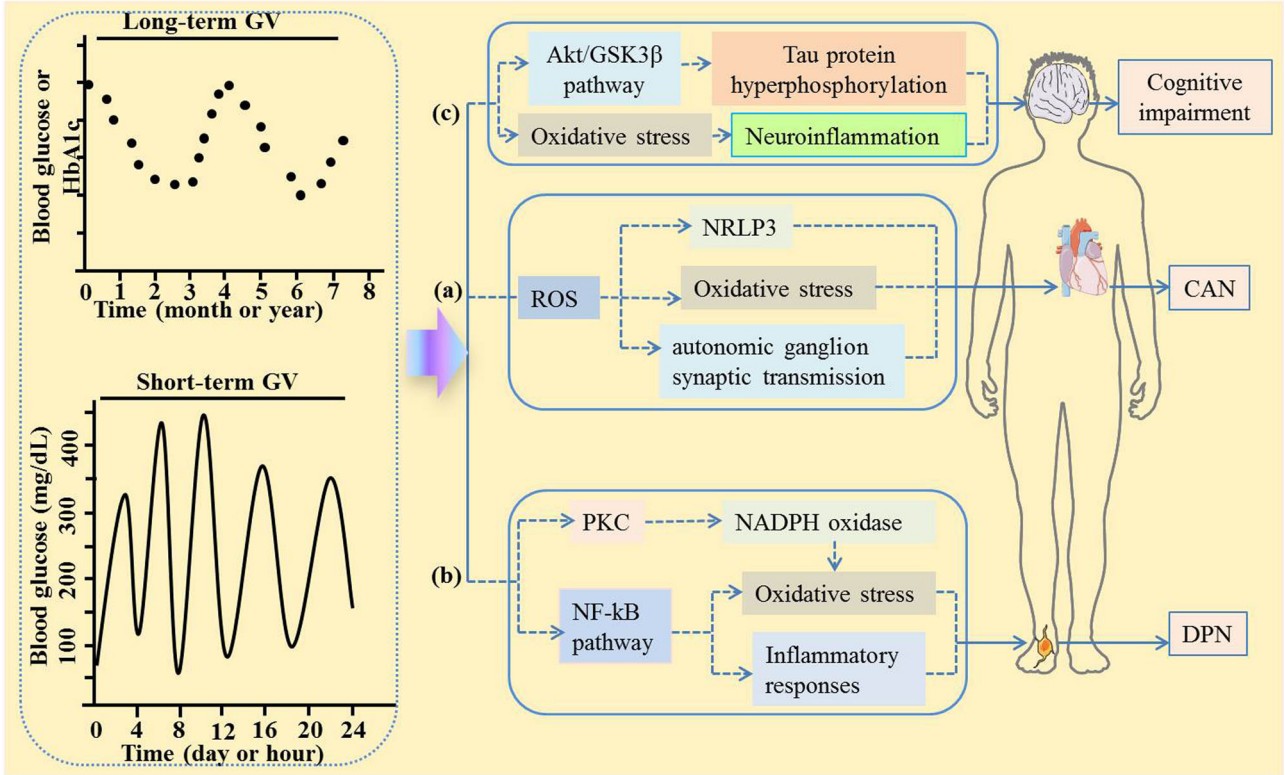

**Fig. 1 Possible mechanism of GV causing DN.** The pathogenesis of DN is complex, and the possible mechanisms can be divided into oxidative stress, inflammatory reactions, etc. The possible mechanisms of GV causing CAN, DPN, and cognitive impairment are as follows: **a** GV increased ROS, which activated the NRLP3 inflammasome and inhibited autonomic ganglion synaptic transmission, thereby leading to CAN; **b** GV induces oxidative stress and inflammatory response by activating the NF-kB pathway or PKC, thereby causing DPN; **c** GV causes cognitive impairment by inhibiting Akt/GSK3β pathway to hyperphosphorylate Tau protein. GV glycemic variability, ROS reactive oxygen species, CAN cardiovascular autonomic neuropathy, DPN diabetic peripheral neuropathy.

**Table 1 Roles of GV in CAN.**

| Metrics of GV | Individuals | Results | References |
|---|---|---|---|
| Low blood glucose index and Under the curve | 44 T1DM patients | Significantly negative correlated with heart rate variability | 49 |
| SD and MAGE | 20 T1DM patients | Correlated with CAN | 50 |
| MAGE | 133 young adults with T1DM | Slightly increase heart rate variability | 51 |
| CV in CGM and all parameters of HbA1c | 110 T2DM patients | Independently associated with CAN | 52 |
| SD of HbA1c | 681 T2DM patients | Significantly associated with CAN | 53 |
| Visit-to-visit HbA1c | 57 T2DM patients | Inversely related to baroreflex sensitivity | 54 |
| Intrapersonal mean, SD, CV for HbA1c | 238 T2DM patients | Strongly associated with CAN | 55 |
| MAGE | 48 men and 39 women with T2DM | Increased GV was associated with CAN in women | 56 |

*CV* coefficient of variation, *SD* standard deviation, *T2DM* type 2 diabetes mellitus, *T1DM* type 1 diabetes mellitus, *HbA1c* hemoglobin A1c, *MAGE* mean amplitude of glycemic excursion, *CAN* cardiovascular autonomic neuropathy.

reactive oxygen species activated the *NRLP3* inflammasome and inhibited autonomic ganglion synaptic transmission by oxidizing the nAch receptor α3 subunit, thereby leading to diabetic CAN[58].

### GV and DPN
DPN is a common chronic complication of long-term diabetes, and its incidence increases with the overall incidence of diabetes. DPN is associated with neuropathic pain, foot ulcers, and subsequent gangrene and amputation, severely affecting the patient's quality of life. In recent years, accumulating evidence has indicated GV as a risk factor of DPN with T1DM. In an 11-year follow-up of 100 patients with T1DM, the standard deviation of blood glucose was proved to be an independent predictor of the prevalence of DPN[1]. In

addition, Kwai et al. found that MAGE was strongly related to excitability markers of altered motor and sensory axonal function, such as super excitability, strength duration time constant, minimum I/V slope etc, indicating that GV may be a key mediator of axonal degeneration as well as a contributing factor in development of DPN with T1DM[61]. Even more importantly, a cross-sectional study found that patients with DPN had higher variability in HbA1c, including HbA1c-SD and HbA1c-CV compared those without DPN[62]. Consistent with this result, a systematic literary review revealed that the increased variability of HbA1c could be used as a biomarker for DPN in foot[63].

Concurrently, multiple cross-sectional studies have shown that the variability of HbA1c is strongly associated with DPN in

patients with T2DM[64–66]. Another observational cohort study enrolled 90 T2DM and with/without DPN, and found that MAGE was significantly correlated with DPN with well-controlled HbA1c[67]. Analogously, Hu et al.[68] enrolled 982 T2DM patients who were screened for DPN and monitored by a continuous glucose monitoring system, and demonstrated MAGE as a significant independent contributor to DPN in type 2 diabetic patients. Furthermore, a retrospective case-control study conducted in Taiwan showed that greater long-term GV was clearly associated with DPN in adults with T2DM[69]. Intriguingly, it was worth to note that GV was also closely linked to the risk of painful DPN in T2DM. A case-control, retrospective study including 275 T2DM with or without painful DPN as well as 351 T2DM without DPN showed that the fasting plasma glucose-CV was significantly correlated with painful DPN risk after multi-variate adjustment[64]. Shortly after, similar results found that increased postprandial glycemic exposure, defined as high HbA1c and near-normal fasting plasma glucose levels, significantly increased the risk of painful DPN in T2DM patients[70]. More recently, Yang et al. revealed that the GV represented by time in range decreased significantly in the mild/moderate/severe pain groups compared with the pain-free group, suggesting that time in range could be used as a valuable clinical evaluation index for painful DPN[71] (Table 2).

As yet, there is little information regarding the effect of GV on DPN. However, it should not be overlooked that there is one of the mechanisms of DPN induced by GV through activating protein kinase C dependent NADPH oxidase, which further lead to oxidative stress[72]. In parallel, GV-induced Schwann cells apoptosis might be involved in this process[73]. Notably, emerging preclinical research showed that GV could weaken the motor nerve conduction velocity of the sciatic nerve, and destroy the microstructure structures of the myelin sheath and axons of the sciatic nerve[74]. In addition, GV significantly reduced the expression of superoxide dismutase, increased the expression levels of malondialdehyde, TNF-a, interleukin-6, and NF-kB. Altogether, studies above indicate that GV induces oxidative stress and inflammatory response by activating the NF-kB pathway, thereby causing DPN (Fig. 1b)[74].

### GV and cognitive impairment

Apart from CAN and DPN, the impact of GV on cognitive impairment such as Alzheimer's disease and vascular dementia has also been addressed[75]. Strikingly, cognitive impairment is twice more frequent in elderly with T2DM. Previous clinical studies had shown that the MAGE was significantly associated with mini-mental status examination, cognition composite score, and brain atrophy in older patients with T2DM[76,77]. Moreover, a cross-sectional study conducted by Kim et al. showed that higher SD or CV of HbA1c was significantly associated with low mini-mental status examination[78]. Furthermore, $Hb_{A1c} \geq 8\%$ in elderly adults with diabetes was related to a worse cognitive ability[79]. Of note, neuroimaging studies had examined the neural relevance of T2DM cognitive impairment and found white matter hyper-intensitie might be the basis of the observed cognitive changes. Indeed, a survey from the Israel Diabetes and Cognitive Decline Study shown that HbA1c variability was significantly associated with APOE4 carrier white matter hyperintensitie volume[80]. Shortly after, a similar result indicated that GV was associated with a higher number of white matter hyperintensitie volume in the multiethnic Washington Heights Inwood Columbia Aging Project[81]. Consistent with these results, Tamura et al. found that high Glycoalbumin/HbA1c, a marker of high GV, was an important determinant factor for large white matter hyper-intensitie volumes in a cross-sectional study[82]. Alongside, AGP showed distinctively different in diabetes-related dementia but not in Alzheimer's disease associated with diabetes, suggesting that diabetes-related dementia group was potentially more sus-ceptible to the deleterious effects of GV on the brain[83] (Table 3). It is important to highlight that patients with T1DM have also been reported to have cognitive impairment[84]. For instance, in a T1DM Exchange Clinic Network including 18 research centers on diabetes, 48% of the participants had clinically significant cog-nitive impairment. In addition, higher HbA1c and continuous glucose monitoring average nocturnal blood glucose were both associated with the increased incidence of clinically significant cognitive impairment[85]. Taken together, all these findings indi-cate that GV may be a contributor to cognitive impairment in diabetic patients.

There is growing evidence that GV significantly drives increased oxidative stress, leading to neuroinflammation and cognitive impairment[86]. With the continuous in-depth research on GV risk factors, neuropathology and neuroimaging provides important mechanism clues for cognitive impairment. Notably, abnormal hyperphosphorylation of Tau protein was thought to play a key role in cognitive impairment[87]. Further support for the idea that GV affecting the risk of cognitive impairment came from another study conducted by Yang et al.[74]. In that study, the authors observed that both learning and memory abilities were disrupted in the fluctuant hyperglycemia rat model, and the mechanism might be that GV inhibited the Akt/GSK3β pathway to hyperphosphorylate Tau protein in the hippocampus, thereby inducing cognitive impairment. Besides, Xia et al. observed that excessive GV was associated with cognitive impairment, as well as significantly reduced degree centrality in the left middle frontal gyrus (Fig. 1c)[88].

### Therapeutic strategies for GV

In light of these above findings, it is time to reconsider the therapeutic strategies for DN. Unfortunately, although several

**Table 2 Roles of GV in DPN.**

| Metrics of GV | Individuals | Results | References |
|---|---|---|---|
| SDBG | 100 T1DM patients | A predictor of the prevalence of DPN | 1 |
| MAGE | 17 T1DM patients | Strongly correlated with excitability markers of DPN | 61 |
| SD and CV of HbA1c | 50 T1DM patients | Long-term GV is associated with DPN | 62 |
| CV-HbA1c, M-HbA1c | 563 T2DM patients | Closely associated with DPN | 65 |
| SD and CV of HbA1c | 223 T2DM patients | Strongly associated with the severity of DPN | 66 |
| SD-blood glucose, MODD, and MAGE | 90 T2DM patients | MAGE is significantly correlating with DPN | 67 |
| MODD, MAGE, and SD | 982 T2DM patients | Significant independent contributor to DPN | 68 |
| Fasting plasma glucose -CV | 2773 T2DM patients | Significantly associated with a risk of painful DPN | 70 |
| Time in range | 364 individuals with DPN | Correlated with painful DPN | 71 |

*DPN diabetic peripheral neuropathy, CV coefficient of variation, SD standard deviation, T2DM type 2 diabetes mellitus, T1DM type 1 diabetes mellitus, HbA1c hemoglobin A1c, MAGE mean amplitude of glycemic excursion, MODD mean of daily differences.*

**Table 3 Roles of GV in cognitive impairment.**

| Metrics of GV | Individuals | Results | References |
|---|---|---|---|
| MAGE | 121 T2DM patients | Significantly correlated with cognitive impairment | 76 |
| Multi-Scale GV | 43 older adults with and 26 without T2DM | Might contribute to brain atrophy and cognitive impairment | 77 |
| SD and CV of visit-to-visit HbA1c | 68 T2DM patients | Visit-to-visit GV influenced cognitive impairment | 78 |
| SD of HbA1c | 124 T2DM patients | Significantly associated with white matter hyperintensitie | 80 |
| Glycoalbumin/HbA1c | 178 elderly patients with diabetes | Independently associated with white matter hyperintensitie | 82 |
| SD and MAGE of HbA1c | 40 patients with AD-related diabetes and 19 patients with diabetes-related dementia | GV is more involved in the pathophysiology of diabetes-related dementia than Alzheimer's disease associated with diabetes | 83 |

CV coefficient of variation, SD standard deviation, T2DM type 2 diabetes mellitus, HbA1c hemoglobin A1c, MAGE mean amplitude of glycemic excursion.

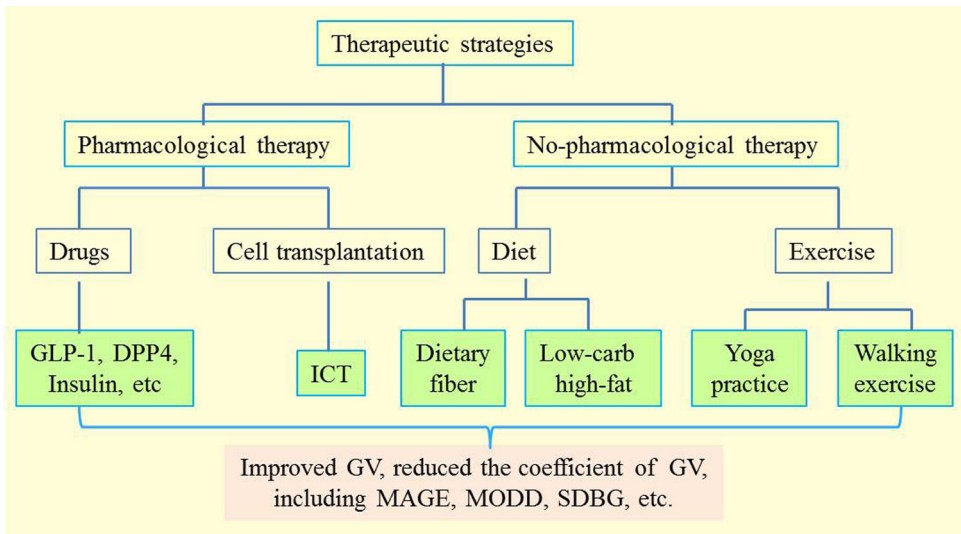

**Fig. 2 Therapeutic strategies to improve GV.** Pharmacological therapy including drugs and insulin as well as non-pharmacological therapy including diet, cell transplantation, and exercise are recommended to improve GV. MAGE mean amplitude of glycemic excursion, MODD mean of daily differences, GLP-1 glucagon-like peptide-1, DPP4 dipeptidyl peptidase 4, ICT islet cell transplantation, SDBG standard deviation of blood glucose.

non-pharmacological treatments of DN have been discussed[89], therapeutic strategies targeting the cause of DN are lacking. Therefore, the recommended pharmacological and non-pharmacological strategies for GV have gained more and more attention (Fig. 2).

### Non-pharmacological therapy

Nutrition therapy is essential in the management of diabetes, where dietary carbohydrate intake is the main factor for post-prandial hyperglycemia and GV. It has been shown that lipid and protein ingested before carbohydrate can significantly improve glucose tolerance by slowing gastric emptying and enhancing insulin secretion[90,91]. Interestingly, Chang et al. found that a very-low-carbohydrate, high-fat breakfast meal could significantly reduce MAGE and SD levels for 24 h, and improve GV in patients with T2DM[92]. Moreover, dietary fiber has shown to reduce postprandial GV in individuals with diabetes. In evaluating the acute effect of fiber-enriched buckwheat and corn pasta on postprandial GV, Vetrani et al.[93] showed that there was a higher stability postprandial GV after fiber-enriched buckwheat pasta in subjects with T1DM. Besides, it is worth noting that use of a moderate amount of sucrose, as part of a balanced diet, does not affect the GV or insulin requirements in T1DM[94]. As such, manipulating the sequence of food intake or choosing more naturally dietary fiber food can improve GV in diabetic patients. In addition, exercise training, including aerobic exercise,

resistance exercise and combined exercise sessions, can also decrease GV in T2DM[95]. In an observational study, Van Dijk et al. showed that prolonged walking exercise could greatly reduce the daily insulin administration in persons with TIDM, but does not necessarily impair 24-h GV[96]. Noteworthily, yoga-assisted treatment of T2DM has multiple benefits, including reduction in fasting plasma glucose, postprandial glucose, oxidative stress, and proinflammatory markers[61]. Similarly, in another study, Vijaya-kumar et al. assessed that there was a significant reduction in GV and a higher duration of time within the glycemic target after one week of yoga practice in T2DM[97].

### Pharmacological therapy

Although different diabetes treatments may reduce HbA1c to similar degrees, their effectiveness in reducing GV may significantly differ[98]. Since hyperglycemia is the main cause of the clinical manifestations and related complications of diabetes, lowering blood glucose levels through pharmacological therapy is the cornerstone of diabetes management.

### Glucagon-like peptide-1 (GLP-1) receptor agonists

The oral hypoglycemic drug metformin is the first-line treatment for T2DM. When blood glucose was poorly controlled in patients with T2DM receiving metformin alone and poorly controlled blood glucose, GLP-1 receptor agonists and basal insulin are used as optional anti-diabetic drugs[99,100]. In contrast to Western

countries, exenatide is a commonly and widely used short-term GLP-1 receptor agonist in China[101]. Exenatide significantly reduces not only standard deviation of the mean blood glucose value and largest amplitude of glycemic excursions, but also highest and mean blood glucose levels[101].

## Dipeptidyl peptidase 4 inhibitors

Vildagliptin is a dipeptidyl peptidase 4 inhibitor that can reduce not only average glycemia but also glucose fluctuation within 24 h by restoring the physiological pattern of insulin and glucagon secretion[102]. Meanwhile, vildagliptin more effectively improved glucose levels with a significantly greater reduction in GV and hypoglycemia than glimepiride in patients with T2DM ongoing metformin therapy[103]. Vianna et al.[104] demonstrated that vildagliptin and gliclazide MR reduced GV(as measured by the MAGE, $p = 0.007$ and 0.034, respectively). Furthermore, an open-label, parallel-group, exploratory study indicated that once-weekly trelagliptin and once-daily alogliptin improved glycemic control and reduced GV without inducing hypoglycemia[105].

## Insulin

Unstable metabolic control and high risk of hypoglycemia due to GV is frequently observed in patients with diabetes on intensive insulin therapy[106]. Thus, the evidence on effectiveness and safety of insulin in patients with diabetes is a priority. Insulin degludec (IDeg), a novel ultra-long-acting basal insulin, has been extensively tested in a comprehensive study involving a wide range of diabetes patients[107–109]. One clinical trial demonstrated that IDeg achieved similar improvements in glycemic control to Iglar in insulin-deficient patients with T2DM, and the day-to-day variation of fasting blood glucose was smaller in patients receiving IDeg[110]. Interestingly, similar findings were reported in another observational longitudinal study[64]. Extensive evidence addresses that IDeg is not only able to reduce GV in patients with T2DM, but also effective in patients with T1DM. Iga et al.[111] found that there were no differences in HbA1c, total insulin dosage, body weight changes, and basal to bolus ratio between the IDeg and IGlar arms. Notably, the day-to-day variability in fasting interstitial GV on the CGM curves was significantly smaller in the IDeg than IGlar treatment period[111].

There has been a large amount of literature on the efficacy and safety of IDeg within basal-bolus regimens in non-hospitalized patients with diabetes, whereas the impact of treatment with IDeg on inpatients has rarely been investigated. An observational longitudinal retrospective study represented that IDeg had the potential to maintain stable levels of blood glucose and reduce GV in hospitalized patients with or without T2DM who require nutritional support[112].

## Cell transplantation

Cell transplantation is being investigated as a possible method of addressing the underlying cause of DN[113,114]. In a pilot clinical study, Mao et al.[115] observed that autologous transplantation of bone marrow mononuclear cells significantly improved the signs and symptoms of DPN. Consistently, another group[116] found that autologous transplantation of bone marrow mononuclear cells improved diabetic sensorimotor polyneuropathy in patients with T2DM, indicating that autologous transplantation of bone marrow mononuclear cells might be an effective and promising treatment for DPN. Of note, islet cell transplantation is another promising treatment for patients with T1DM and severe hypoglycemia that is resistant to other therapies[117]. Islet cell transplantation may reduce complications through both improved glycemic control and reduction in GV[117,118]. The results of one prospective, crossover study demonstrated that islet cell transplantation could slow the progression of diabetic retinopathy and nephropathy compared with intensive medical therapy[119]. In addition, Azmi et al. found that HbA1c, neuropathy symptoms and peroneal nerve conduction velocity were improved in T1DM patients after simultaneous pancreas and kidney transplantation[120]. However, additional clinical studies will be required to confirm the safety and efficacy of cell transplantation in the treatment of DN.

## Combination therapy

Misra et al. reported a young woman with a KCNJ11-G334V mutation who showed significant improvements in glycaemic control when treated with high-dose sulfonylureas therapy combined with insulin[121]. Notably, this combination therapy also resulted in marked improvements in GV and hypoglycemic awareness. Similarly, a proof-of-concept study confirmed efficacy and safety of mealtime exenatide treatment for a high-risk insulin-requiring population and demonstrated a reduction of GV using this approach[98]. A double-blind randomized phase 2 study demonstrated that empagliflozin significantly reduced GV and increased time spent in the glucose targetrange without increasing time spent in hypoglycemia[122]. Nomoto et al. proposed that combination therapy of dapagliflozin and insulin injection did not show glucose fluctuation superiority over dipeptidyl peptidase 4 inhibitors on insulin therapy[123]. If reduced glycemic excursions are the treatment priority, a regimen using GLP-1 is preferable to basal insulin alone because of the shorter time above range and reduced GV[124]. Particularly, GV decreased only when GLP-1 was a part of the treatment regimen[124]. Nevertheless, in a prospective substudy of the Qatar Study, Ponirakis et al. found that treatment with exenatide plus pioglitazone or insulin reduced HbA1c and promoted small fiber regeneration, but had no impact on neuropathic pain over 1 year[125].

Strikingly, a recent study reported that administration of hyocholic acid in diabetic mouse could improve fasting GLP-1 secretion and glucose homeostasis. Subsequently, in a clinical cohort, it was further confirmed that low concentration of hyocholic acid in serum was related to diabetes[126]. Thus, hyocholic acid is expected to be developed as another new drug for the treatment of diabetes GV, thereby improving DN. In addition, in terms of blood glucose control, the aim of treatment for T2DM is to reduce HbA1c to the target level and reduce GV in order to avoid both hypoglycemia and wide fluctuations of postprandial glucose[127]. Therefore, DN can be slowed down by early detection of autonomic imbalance, attention to diabetes control, and elimination of risk factors for neuropathy[48].

## Summary and further perspectives

Currently, assessment of GV in routine clinical practice remains a challenge. Indeed, there is no gold standard for evaluation of GV. Therefore, it becomes essential for clinical practice to adopt the best methods available for the evaluation of GV to provide the most relevant feedback to improve glycemic control[128]. Moreover, patients with diabetes face a life-long optimization problem of how to avoid hypoglycemia as well as lower average blood sugar levels and postprandial hyperglycemia[16]. This optimization can be achieved if GV is reduced. Consequently, in the past decade, along with HbA1c, GV has been increasingly regarded as a primary marker of glycaemic control[17–19]. However, for the same plasma insulin concentrations, hypoglycemic effects may differ, depending on insulin sensitivity, even after intravenous administration. In a relatively short period of time, insulin sensitivity varies considerably within and between individuals, leading to different metabolic effects in patients with diabetes[106]. Therefore, no-pharmacological therapy has become an effective way to improve blood GV. Altogether, DN is one of the most

common long-term complications of diabetes, and good GV control may be essential in prevention of such complications.

**Reporting summary**. Further information on research design is available in the Nature Research Reporting Summary linked to this article.

## Data availability
There were no data used in this study and thus no data are available.

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

## Acknowledgements

This work was supported by funding from the National Natural Science Foundation of China Grant 82104307 (to S.B.) and the Natural Science Foundation of Hunan Province Grant 2021JJ40865 (to S.B.).

## Author contributions

Z.X.C. and Y.X. contributed towards the concept and manuscript writing; Z.C.S. and S.B. revised and supervised overall project.

## Competing interests

The authors declare no competing interests.
