## [Transparent Peer Review File · Communications Biology]

Reviewers' comments:

Reviewer #1 (Remarks to the Author):

This is a timely and novel review of the relationship between glucose variability assessed using CGM with diabetic somatic and autonomic neuropathy.

1. For the effect of GV on progression of DN consider the study by Ishibashi et al Diabetes Care. 2019 Jan;42(1):110-118.
2. A correlation between GV and HRV-DB cannot be concluded as GV driving CAN, but the opposite that patients with CAN have higher GV.
3. Line 149- What is 'diabetic foot peripheral neuropathy'?
4. On Lines 175-178 the authors state: 'Notably, emerging preclinical research showed that GV could weaken the motor nerve conduction velocity of the sciatic nerve, and the microstructure structures of the myelin sheath and axons of the sciatic nerve were destroyed'. There is no reference to this study.
5. On lines 196-198 the authors state 'Shortly after, similar results were found in the multiethnic Washington Heights Inwood Columbia Aging Project'. What results?
6. Line 234: 'very low carbohydrate high-fat' What?
7. With regard to transplantation the authors have not considered the reported benefit of simultaneous pancreas and kidney transplantation (Azmi et al Diabetologia. 2019 Aug;62(8):1478-1487).
8. In section 4.2.5 the authors suggest that 'If reduced glycemic excursions are the treatment priority, a regimen using GLP-1 is preferable to basal insulin'. However, a recent study has shown a comparable effect of GLP-1 with basal bolus insulin on DPN outcomes (Ponirakis et al BMJ Open Diabetes Res Care. 2020 Jun;8(1):e001420).
9. What is the relevance of aldose reductase inhibitors? Have they been shown to reduce GV?
10. The final paragraph on sex-dimorphic effects on insulin secretion in the summary and perspective section is completely out of place and irrelevant in the context of the review being on GV and DPN.

Reviewer #2 (Remarks to the Author):

This is a perspective review focusing on the role of glycemic variability (GV) on the development of diabetic neuropathy. Overall, the manuscript was written well and easy to follow. There are some comments.

1. The authors picked up both short-term and long-term GV, but correlation between these indices is not clearly discussed. The authors should discuss the similarity and differences between short-term and long-term GV, as well as among various indices of short-term GV. The authors may refer to the following articles: Tsuchiya et al. Endocr J 67:877-881, 2020, Saisho et al. Prim Care Diabetes 9:290-296, 2015, Saisho. Int J Mol Sci 15:18381-406, 2014.
2. Figure 1. Cell transplantation should be classified as pharmacotherapy. The information in this figure was not comprehensive and only limited to some specific drugs or therapies. This figure should be revised in a more comprehensive and self-explanatory manner, or removed.
3. Figure 2. This figure implies that the mechanisms of development of neuropathy were different among CAN, DPN and dementia. To avoid confusion, the common and specific mechanisms of CAN, DPN and dementia should be described separately.
4. There are some grammatical errors throughout the manuscript. The manuscript should be edited by a native English speaker.

Reviewer #1 (Remarks to the Author):

This is a timely and novel review of the relationship between glucose variability assessed using CGM with diabetic somatic and autonomic neuropathy.

1. For the effect of GV on progression of DN consider the study by Ishibashi et al Diabetes Care. 2019 Jan;42(1):110-118.

Reply: Thank you very much for your valuable suggestions, it will be very helpful to improve the quality of our article. We have referred to this document and enriched this part of the content. See line 25-42 for details.

2. A correlation between GV and HRV-DB cannot be concluded as GV driving CAN, but the opposite that patients with CAN have higher GV.

Reply: Thank you very much. Indeed, we have not clarified the correlation between GV and CAN in this section, and we modified the corresponding part in the revised manuscript. Based on several literature reports, we mainly explained that GV was an important risk factor leading to CAN in the section 3.1.

3. Line 149- What is ‘diabetic foot peripheral neuropathy’?

Reply: We refer to the literature of Casadei G et al, Glycated Hemoglobin (HbA1c) as a Biomarker for Diabetic Foot Peripheral Neuropathy[J] .Diseases, 2021, 9(1): 16. ‘diabetic foot peripheral neuropathy’ mentioned in this article refers to the manifestation of diabetic peripheral neuropathy in foot. In order to avoid ambiguity, we have made corresponding changes in the revised manuscript. See line 173-174 for details.

4. On Lines 175-178 the authors state: ‘Notably, emerging preclinical research showed that GV could weaken the motor nerve conduction velocity of the sciatic nerve, and the microstructure structures of the myelin sheath and axons of the sciatic nerve were destroyed’. There is no reference to this study.

Reply: Thank you for your advice. According to your advice, we have added the references to this study. See line 204 for details.

5. On lines 196-198 the authors state ‘Shortly after, similar results were found in the multiethnic Washington Heights Inwood Columbia Aging Project’. What results?

Reply: We have enriched the content of this part. See line 224-225 for details.

6. Line 234: ‘very low carbohydrate high-fat’ What?

Reply: Thank you for your advice. We have modified ‘very low carbohydrate high-fat’ to ‘a very-low-carbohydrate, high-fat breakfast meal’ in the revised manuscript. See line 268-269 for details.

7. With regard to transplantation the authors have not considered the reported

benefit of simultaneous pancreas and kidney transplantation (Azmi et al Diabetologia. 2019 Aug;62(8):1478-1487).

Reply: Thank you for your suggestions. According to your advice, we had incorporated this literature report into the article.

8. In section 4.2.5 the authors suggest that ‘If reduced glycemc excursions are the treatment priority, a regimen using GLP-1 is preferable to basal insulin’. However, a recent study has shown a comparable effect of GLP-1 with basal bolus insulin on DPN outcomes (Ponirakis et al BMJ Open Diabetes Res Care. 2020 Jun;8(1):e001420).

Reply: Thank you very much for your valuable suggestions, it will be very helpful to improve the quality of our article, we had incorporated this literature report into the article. See line 349-351 for details.

9. What is the relevance of aldose reductase inhibitors? Have they been shown to reduce GV?

Reply: Indeed, considering that there was no related reports on aldose reductase inhibitors and GV, we have deleted this part of the text.

10. The final paragraph on sex-dimorphic effects on insulin secretion in the summary and perspective section is completely out of place and irrelevant in the context of the review being on GV and DPN.

Reply: Thank you for your suggestion, we have removed this part of the description in the revised manuscript.

Reviewer #2 (Remarks to the Author):

This is a perspective review focusing on the role of glycemc variability (GV) on the development of diabetic neuropathy. Overall, the manuscript was written well and easy to follow. There are some comments.

1. The authors picked up both short-term and long-term GV, but correlation between these indices is not clearly discussed. The authors should discuss the similarity and differences between short-term and long-term GV, as well as among various indices of short-term GV. The authors may refer to the following articles: Tsuchiya et al. Endocr J 67:877-881, 2020, Saisho et al. Prim Care Diabetes 9:290-296, 2015, Saisho. Int J Mol Sci 15:18381-406, 2014.

Reply: Thank you for your suggestion, it will be very helpful to improve the quality of our article. We discussed the similarity and differences between short-term and long-term GV, as well as among various indices of short-term GV (see line84-91 for details).

2. Figure 1. Cell transplantation should be classified as pharmacotherapy. The information in this figure was not comprehensive and only limited to some

specific drugs or therapies. This figure should be revised in a more comprehensive and self-explanatory manner, or removed.

Reply: Thank you for your comments. According to your advice, we have modified the Figure 1 in the revised manuscript.

3. Figure 2. This figure implies that the mechanisms of development of neuropathy were different among CAN, DPN and dementia. To avoid confusion, the common and specific mechanisms of CAN, DPN and dementia should be described separately.

Reply: Thank you very much. In response to this issue, we have supplemented the common and specific mechanisms of CAN, DPN and dementia in Figure 2. See line828-831 for details.

4. There are some grammatical errors throughout the manuscript. The manuscript should be edited by a native English speaker.

Reply: Thank you for your suggestion. The grammar mistakes have been checked and corrected by the native English speaker throughout the manuscript.

REVIEWERS' COMMENTS:

Reviewer #1 (Remarks to the Author):

The authors have addressed my major concerns.

Reviewer #2 (Remarks to the Author):

The authors have responded to the comments appropriately.

Reviewer #1 (Remarks to the Author):

The authors have addressed my major concerns.

Reply: Thank you very much for your valuable suggestion, it will be very helpful to improve the quality of our article.

Reviewer #2 (Remarks to the Author):

The authors have responded to the comments appropriately.

Reply: Thank you again for your valuable suggestion, which is very helpful to improve and enrich our articles.